# Rational Design of an Ion-Imprinted Polymer for Aqueous Methylmercury Sorption

**DOI:** 10.3390/nano10122541

**Published:** 2020-12-17

**Authors:** Ruddy L. Mesa, Javier E. L. Villa, Sabir Khan, Rafaella R. Alves Peixoto, Marcelo A. Morgano, Luís Moreira Gonçalves, Maria D. P. T. Sotomayor, Gino Picasso

**Affiliations:** 1Laboratory of Physical Chemistry Research, Faculty of Sciences, National University of Engineering, Lima 15333, Peru; a20156475@pucp.edu.pe (R.L.M.M.); sabir_chemist@yahoo.com (S.K.); 2Institute of Chemistry, State University of São Paulo (UNESP), Araraquara, SP 14800-060, Brazil; villa.javier03@gmail.com; 3National Institute for Alternative Technologies of Detection, Toxicological Evaluation and Removal of Micropollutants and Radioactives (INCT-DATREM), Araraquara, SP 14800-060, Brazil; 4Department of Analytical Chemistry, Fluminense Federal University (UFF), Niterói, RJ 24020-150, Brazil; rafaellapeixoto@id.uff.br; 5Institute of Food Technology (ITAL), Campinas, SP 13070-178, Brazil; morgano.ital@gmail.com; 6Institute of Chemistry, University of São Paulo (USP), São Paulo, SP 05508-000, Brazil; lmgoncalves@iq.usp.br

**Keywords:** bulk polymerization, computational modelling, environmental analysis, imprinting technology, mercury detection and removal, ion recognition, ionic imprinting polymers, sample preparation, separation science, water analysis

## Abstract

Methylmercury (MeHg^+^) is a mercury species that is very toxic for humans, and its monitoring and sorption from environmental samples of water are a public health concern. In this work, a combination of theory and experiment was used to rationally synthesize an ion-imprinted polymer (IIP) with the aim of the extraction of MeHg^+^ from samples of water. Interactions among MeHg^+^ and possible reaction components in the pre-polymerization stage were studied by computational simulation using density functional theory. Accordingly, 2-mercaptobenzimidazole (MBI) and 2-mercaptobenzothiazole (MBT), acrylic acid (AA) and ethanol were predicted as excellent sulfhydryl ligands, a functional monomer and porogenic solvent, respectively. Characterization studies by scanning electron microscopy (SEM) and Brunauer–Emmett–Teller (BET) revealed the obtention of porous materials with specific surface areas of 11 m^2^ g^−1^ (IIP–MBI–AA) and 5.3 m^2^ g^−1^ (IIP–MBT–AA). Under optimized conditions, the maximum adsorption capacities were 157 µg g^−1^ (for IIP–MBI–AA) and 457 µg g^−1^ (for IIP–MBT–AA). The IIP–MBT–AA was selected for further experiments and application, and the selectivity coefficients were MeHg^+^/Hg^2+^ (0.86), MeHg^+^/Cd^2+^ (260), MeHg^+^/Pb^2+^ (288) and MeHg^+^/Zn^2+^ (1510), highlighting the material’s high affinity for MeHg^+^. The IIP was successfully applied to the sorption of MeHg^+^ in river and tap water samples at environmentally relevant concentrations.

## 1. Introduction

Methylmercury (MeHg^+^) is one of the mercury chemical forms of highest toxicity for humans among mercury compounds [1]. Once in the human body, MeHg^+^ presents a high affinity to the sulfhydryl group present in proteins. This interaction can induce changes in the protein structures and a consequent loss of their functions [2]. The deleterious effects are more severe for fetuses; MeHg^+^ exposure can seriously impair neurological development [3]. In addition, MeHg^+^ is bioaccumulated and biomagnified in the marine food chain, as this specie is found in almost all marine species as a result of methylation of inorganic mercury by microorganisms present in sediments and aquatic organisms [4]. In natural water, reports estimated that MeHg^+^ percentage varied from 0.27 to 20% in relation to total mercury [5,6]. In this way, the consumption of contaminated water and seafood is the main source of non-occupational exposure to MeHg^+^. For these reasons, the development of analytical strategies with the aim of MeHg^+^ determination/removal is a question of public health concern. Although several methods have been proposed for the determination and removal of MeHg^+^ in various samples [7,8,9,10,11,12,13], the synthesis and application of highly selective materials for efficient determination/removal of MeHg^+^ is a topic that deserves further attention.

Imprinted polymers are materials with the capacity to selectively capture molecules or metal ions and have become very popular sorbents for a wide variety of applications [14,15,16]. Particularly, ion-imprinted polymers (IIPs) are sorbents designed for targeting metal ions and have been recently used as solid phase extraction materials, sensors and separating membranes [17,18,19,20,21,22]. In the synthesis of IIPs, ligands containing electron-rich heteroatoms, such as nitrogen, phosphorus, sulfur and oxygen, are normally used to promote the formation of chelate with the targeted metal [23]. This three-step synthesis normally involves: (1) formation of a pre-polymerization complex with interaction of the metal ion, ligand and functional monomer (FM); (2) polymerization of the complex monomer with the addition of a cross-linker and initiator; (3) removal of the template to produce the imprinted polymer [24]. After proper removal of the metal ions from the bulk, specific cavities/binding sites able to selectively recognize the metal are generated [25].

IIPs for MeHg^+^ commonly require a trapping technique using a pre-polymerization mixture based on a ternary template/ligand/FM. The template is normally the analyte, using methylmercury chloride (MeHgCl) [26,27,28,29,30] as the ion source, although MeHgCl is a hardly soluble salt. Ligands containing sulfhydryl groups are the more suitable ligands since mercury ions have a specific affinity to sulfur, even though ligands containing *N*-based functional groups have been found to interact with MeHg^+^ [24]. 1-pyrrolidinecarbodithioic acid [26,27,28], phenobarbital [29,30,31] and methacrylic acid [26,27,28,29,30,31] have been reported in the synthesis of MeHg^+^ IIPs such as sulfhydryl ligand, *N*-based ligand and FM, respectively. In the present study, sulfhydryl ligands (2-mercaptobenzimidazole and 2-mercaptobenzothiazole) are tested for first time in MeHg^+^ IIP synthesis.

Computational simulation allows the evaluation of several components for IIP synthesis and their interaction; it is assumed that the formed complexes preserve their chemical structures after polymerization, thus, only the pre-polymerization system is modeled [32,33]. Therefore, the consideration of individual interactions among template, ligand, FM, cross-linker and solvent might significantly reduce the time and resources to choose the components for IIP synthesis. Density functional theory (DFT) methods are typically used to calculate the “binding energy” for polymer components selection [34,35].

In this work, theoretical calculations were used to predict relevant experimental conditions in the synthesis of an IIP for aqueous MeHg^+^ sorption. DFT was used for geometry optimization and binding energy calculations in systems containing MeHg^+^, sulfhydryl ligands, FM and solvent. For the validation of the theoretical results, IIPs and non-imprinted polymers (NIPs) were synthesized by bulk polymerization and characterized by Fourier transform infrared spectroscopy (FTIR), scanning electron microscopy (SEM), thermogravimetric analysis (TGA) and Brunauer–Emmett–Teller (BET) measurements. The sorption and selectivity study were performed to assess the efficiency of the proposed MeHg^+^–IIP and for comparison with the corresponding NIPs. Finally, the MeHg^+^–IIP was successfully applied to the quantitative sorption of MeHg^+^ in samples in river and tap water at environmentally relevant concentrations.

## 2. Materials and Methods

### 2.1. Reagents and Samples

MeHgCl, tiourea, 2-mercaptobenzimidazole (MBI), 2-mercaptobenzothiazole (MBT), acrylic acid (AA), ethylene glycol dimethacrylate (EGDMA) and azodiisobutyronitrile (AIBN) were purchased from Sigma-Aldrich (St. Louis, MO, USA); ethanol (EtOH) absolute (99.9%) was obtained from JT Baker (Radnor, PA, USA); MeHgCl standard solution 1000 mg L^−1^ from Alfa Aesar (Tewksbury, MA, USA); and hydrochloric acid (HCl), sodium acetate (NaCH_3_COO), acetic acid (CH_3_COOH), sodium hydrogen phosphate (Na_2_HPO_4_), sodium dihydrogen phosphate (NaH_2_PO_4_), sodium hydroxide (NaOH), and sodium tetraborate (Na_2_B_4_O_7_) were purchased from Merck (Darmstadt, Germany);. For the selectivity study, solutions were prepared from salt standards of lead nitrate, cadmium nitrate tetrahydrate, zinc sulfate monohydrate, and mercury dichloride (all acquired form Sigma-Aldrich). All the reagents used in this work were of analytical grade, and the solutions were prepared in ultrapure water (resistivity of 18.2 MΩ cm).

The samples were collected from the Batalha river (21.72 S, 49.20 W), and directly from the tap.

WARNING: MeHgCl appears as white microcrystals and is very toxic by inhalation, in contact with skin and if swallowed. Protection measures such as glasses, full face-shields, gloves and particulate respirators are required to manipulate it. Its disposal should also be carried out carefully.

### 2.2. Density Functional Method

DFT is a computational method applied to electronic systems to obtain the energy and the electronic distribution of the fundamental state, which is a simpler magnitude than the wave function and therefore simplifies the study of complex systems [33,34]. Accordingly, the binding energies were calculated using Gibbs free energies (Δ*G*), which is a compressive parameter (involving enthalpy, entropy and temperature) and a determinant thermodynamic criterion for a spontaneous process [32]. The computational simulations were performed using a Gaussian 09 package [36], M06-2X level of theory [37] and basis set 6-31G(d) [38,39] for the carbon, sulfur, nitrogen, oxygen and hydrogen atoms. In the case of MeHg^+^, a pseudopotential (LANL2DZ) [40] was employed to describe the core orbitals of the mercury atom.

### 2.3. Ion-Imprinted Polymer (IIP) Synthesis

MeHg^+^–IIPs were synthetized via bulk polymerization using MeHgCl (0.5 mmol) as the template; MBI (0.5 mmol) or MBT (0.5 mmol) as ligands for comparison; AA (1.0 mmol) as FM; EGDMA (2 mmol) as a cross-linker, AIBN (0.5 mmol) as radical initiator; EtOH as solvent (20 mL). First, the template and ligand were dissolved in EtOH, the FM was added, and the mixture was kept under vigorous stirring for 2 h. Subsequently, the cross-linker and the radical initiator were added under nitrogen atmosphere. Then, the glass tube was sealed under an inert environment, and the temperature was set to 65 °C to initiate the polymerization. A schematic illustration of the IIP synthesis is shown in Figure 1.

The template ion MeHg^+^ in the IIPs was removed with a mixture of tiourea 2.5% *w/v* and HCl 1 mol L^−1^ solution until mercury could no longer be detected in the solution by thermal decomposition amalgamation atomic absorption spectrometry (TDA–AAS), following a previously proposed method [10]. The polymer was dried at 60 °C and stored at room temperature. Two non-imprinted polymers (NIPs) were synthesized following the same procedure, but no template was added in one of them, and neither template nor ligand was added in the other. Thus, the synthesized materials were IIP–MBI–AA and IIP–MBT–AA (CH_3_Hg^+^ imprinted polymers), NIP–MBI–AA and NIP–MBT–AA (non-imprinted polymers with ligand) and NIP–AA (non-imprinted polymers without ligand).

### 2.4. Polymer Characterization

Characterizations of the MeHg^+^–IIPs and NIPs were carried out by attenuated total reflection (ATR)–FTIR in a Bruker spectrometer (Billerica, MA, USA) Alpha II in the range of 4000–400 cm^−1^ to identify functional groups in each material. Morphology was examined by scanning electron microscopy (SEM) of the JEOL brand model JSM7500F (Tokyo, Japan). The samples were dried at 60 °C, placed on a carbon wafer and then subjected to a sputtering using a spray coating equipment of the Bal Tec brand model SCD 050. The nitrogen adsorption/desorption analysis was used to evaluate the surface area by the BET method (Brunauer–Emmett–Teller), whereas the total volume and pore diameter was evaluated by the BJH method (Barret–Joyner–Halenda) using a Micromeritics Gemini VII-2390 (Norcross, GA, USA). The samples were pretreated in a vacuum by degassing at 100 °C for 2 h. The thermogravimetric analysis (TGA) and differential thermal analysis (DTA) were applied to evaluate the thermal stability of the materials. Instrument Perkin Elmer STA 6000 (Waltham, MA, USA) was used to obtain the thermograms. The thermograms were obtained using 5 mg of mass of the sample and a heating ramp of 10 °C min^−1^ in the range of from 35 to 800 °C in nitrogen atmosphere with a flow rate of 20 mL min^−1^ using a Perkin Elmer STA 6000 instrument.

### 2.5. Sorption Studies

Sorption experiments of MeHg^+^ from aqueous solutions were carried out by TDA–AAS, using a direct mercury analyzer (DMA) for the determination of total mercury [10]. Sorption isotherms were studied at different MeHg^+^ concentrations (25, 50, 100, 150, 200, 300, 400, 600 and 800 mg L^−1^). Polymer concentration was kept constant at 3 mg of polymeric material in 2 mL of MeHg^+^ solution and shaken for 2 h. Then, the polymers were separated by centrifugation for 10 min at 450 RCF. After that, the concentration of MeHg^+^ was quantified in the supernatant using TDA–AAS. The effect of pH on MeHg^+^ sorption was also studied with the following buffer solutions: HCl for pH 2.0; NaCH_3_COO/CH_3_COOH for pH 4.0–6.0; Na_2_HPO_4_/NaH_2_PO_4_ for pH 7.0–8.0; Na_2_B_4_O_7_/NaOH for pH 10.

The sorption kinetic was studied in the range of 5–300 min. Afterwards, the samples were centrifuged for 10 min at 450 RCF, and the residual solution was analyzed by TDA–AAS. The other parameters were kept constant (100 µg L^−1^ MeHg^+^ solution, pH 8 with Na_2_HPO_4_/NaH_2_PO_4_ buffer, polymer concentration: 3 mg of polymeric material in 2 mL MeHg^+^ solutions, and 2 h of stirring time).

### 2.6. Selectivity Studies and Water Analysis

The selectivity of the synthesized polymer for MeHg^+^ was assessed in binary mixtures of MeHg^+^/Hg^2+^, MeHg^+^/Cd^2+^, MeHg^+^/Pb^2+^ and MeHg^+^/Zn^2+^. A polymer mass of 3 mg and 2 mL standard MeHg^+^ solutions at pH 8 were used for these experiments under shaking for 2 h. The mixture was centrifuged for 10 min at 450 RCF, and the concentration of metal ions was quantified using inductively coupled plasma with optical emission spectrometry (ICP-OES; Optima 3300 DV from Perkin Elmer). The same procedure was used to analyze the water samples. To evaluate the application of the materials, the two water samples were analyzed with the same procedure. These were filtered using Whatman #42 paper and tested by spiking and recovery experiments. The MeHg^+^ in the IIP was removed with a mixture of tiourea, 2.5% *w/v*, and 1 mol L^−1^ of HCl following a previously proposed methodology [27], and it was then quantified by TDA–AAS [10].

## 3. Results and Discussion

### 3.1. Theoretical Selection of the Functional Monomer and Solvent

A methodology based on the geometry optimization was employed to obtain the lowest energy conformation followed by the calculation of the binding energy (Δ*E*_1_) of each metal ion–ligand complex. This process was also carried out for the metal ion–ligand–FM complex (Δ*E*_2_). The interaction energies of complexes were based on the following equations (where *n* is number of FM for metal ion–ligand–FM complex calculations):(1)$$\Delta E_{1}=E\left(metal\;ion-ligand\right)-E\left(metal\;ion\right)-E\left(ligand\right)\;,\;$$
(2)$$\;\Delta E_{2}=E\left(metal\;ion-ligand-FM\right)-E\left(metal\;ion-ligand\right)-nE\left(FM\right)\;,\;$$

The assessment of these energies in different solvents (water, EtOH, dimethyl sulfoxide (DMSO), dimethylformamide (DMF) and acetonitrile) was also carried out aiming to predict the most stable complex.

The geometry optimization was performed for MeHg^+^ complexes with the sulfhydryl ligands 2-mercaptobenzimidazole (MBI) and 2-mercaptobenzothiazole (MBT). As the linear coordination is the most likely geometry for MeHg^+^ complexes [41], the formation of linear complexes of MeHg^+^–MBI and MeHg^+^–MBT was considered in the geometry optimization in EtOH medium (see Figure 2). For theoretical calculations, the Hg–S distances were calculated to be 2.717 Å and 2.698 Å for MeHg^+^–MBT and MeHg^+^–MBI, respectively, which were similar to the experimental values previously reported [42]. Some additional theoretical and experimental bond lengths for these complexes can be found in Appendix A.

The effect of the solvent on the optimized geometries was also considered, and the binding energies for the MeHg^+^ complexes are presented in Table 1. The results display negative value of *ΔE*, which indicates spontaneous complex formation and an effective interaction between the Hg atom and sulfhydryl groups. The most stable (lowest) values obtained were −15.4 kcal mol^−1^ for MeHg^+^–MBI and −12 kcal mol^−1^ for MeHg^+^–MBT in EtOH, and it was therefore selected as the porogenic solvent.

Regarding the FM, it should possess higher binding energy toward the ligand than the binding energy with the template to allow for the template/analyte to be adsorbed and desorbed from the IIP bulk. Figure 2 and Appendix A show the chemical structures and the *ΔE* of the pre-polymerization stage in the EtOH solvent. These results show that AA provides stronger IIPs than basic and neutral monomers. MeHg^+^ interacts with MBI (−13.3 kcal mol^−1^) and MBT (−11 kcal mol^−1^) by coordinate bonds (1:1), and AA interacts with MBI (−20 kcal mol^−1^) and MBT (−12.2 kcal mol^−1^) by hydrogen bonds. Thus, after the theoretical selection of two potentially efficient ligands (MBI and MBT), the FM (AA) and solvent (EtOH), these results were experimentally validated by synthesizing and comparing of the performances of the IIP–MBI–AA and IIP–MBT–AA materials.

### 3.2. Characterization of the Synthesized Polymers

Appendix A shows the comparison of the FTIR spectra of all the synthesized polymers, which presented similar spectral profiles. IIPs and NIPs showed characteristic peaks of C=O stretch (1724 cm^−1^), C–O stretch (1146 cm^−1^), C–O bend (1253 cm^−1^), C–H stretch (2980 cm^−1^) [43], CH_3_ and CH_2_ bends (1392 and 1452 cm^−1^, respectively). As expected, no band is present in the region of 1645–1630 cm^−1^ of C=C groups in polymer materials, suggesting the complete polymerization of EGDMA. The sulfhydryl ligands MBI and MBT were also characterized by FTIR; however, the characteristic peaks associated with MBI (N–C=S stretch (1174 cm^−1^) and S–H stretch (2877 cm^−1^)) and with MBT (NS–H stretch (2890 cm^−1^), N–C=S stretch (1082 cm^−1^) and C–S stretch (748–667 cm^−1^)) were not observed [44,45]. It should be noted that, the FTIR spectroscopy analysis was not able to provide vibrational information about the ligands in the polymer network because of the small amount of MBI or MBT used in the reaction compared to the cross-linker (EGDMA).

The surface morphology of the synthesized materials was studied by SEM analysis, and the results are shown in Figure 3. In all the cases, rounded particles were obtained, and a slightly higher roughness for the IIP–MBT–AA and NIP–MBT–AA materials was observed. As there were no clear difference among the morphology of the synthesized polymers by SEM analysis, the quantitative measurement of the effective surface area was necessary.

Nitrogen adsorption/desorption analysis was carried out to estimate the pore surface area by BET, and the pore size and total pore volume by Barret–Joyner–Halenda (BJH) analysis. The results are presented in the Table 2, and it can be observed that IIP–MBI–AA presented the largest surface area (11 m^2^ g^−1^) and the smallest pore size (9.5 nm). This suggests the presence of more imprinted sites for MeHg^+^ and a high adsorption capacity for the IIP–MBI–AA polymer. The IIP–MBT–AA and NIP–MBT–AA materials displayed similar surface areas (5.3 and 5.5 m^2^ g^−1^, respectively) and pore diameter (11 and 12 nm, respectively), which indicates a small difference on the sorption capacity. Based on this data, IIP–MBI–AA might be considered as a better sorbent than IIP–MBT–AA since the MBI ligand interacts with two hydrogen bonds in the polymer network, creating more binding sites to ensure migration of MeHg^+^. On the other hand, the MBT ligand only interacts with one hydrogen bond, indicating lower porosity towards binding sites. However, quantitative sorption studies are necessary to corroborate these hypotheses.

Thermal stability of the polymers was evaluated by TGA and DTA techniques, and the results can be found in Appendix A. All polymers showed similar degradation pattern. The first endothermic event is observed due to the loss of intrinsically bound water at a temperature below 100 °C in the TGA/DTA curve. This peak is followed by another endothermic peak which is responsible for the main decomposition of the polymers (282–333 °C). It could be that this is attributed to the short chain degradations as well as decarboxylation process, and was continued until 400 °C [46,47,48]. Nearly 100% of total mass of the polymers was decomposed in the range of 282–400 °C. This means that all polymers possess good thermal stability, tolerating temperatures until 282 °C without significant degradation.

### 3.3. Adsorption Studies

The effect of pH on MeHg^+^ sorption towards IIPs and NIPs was studied in the pH range from 2 to 10, and the results are presented in Figure 4A. In all materials, the sorption capacity increased rapidly and reached the maximum capacity at pH 8. For IIP–MBI–AA, the maximum value was 39 µg g^−1^, whereas for NIP–MBI–AA and NIP–AA, the maximum were 30 µg g^−1^ and 17 µg g^−1^, respectively. On the other hand, IIP–MBT–AA (73 µg g^−1^) and NIP–MBT–AA (70 µg g^−1^) display similar adsorption capacity. These results indicate that despite the fact that the IIP–MBI–AA polymer displayed the larger surface area, the IIP–MBT–AA polymer exhibited a highest adsorption capacity. This was probably due to the larger pore size of IIP–MBT–AA which facilitates the analyte sorption. The pH has a strong influence on MeHg^+^ speciation; for example, a pH around 6–8 dictates that most of the MeHg^+^ would have a hydroxide group associated with it, while in low pH 2–4, ionic speciation for MeHg^+^ is present. pH 8 was selected to perform all the following experiments because the sorption was more efficient at this pH value. The same trend was observed in another IIP for MeHg^+^ where the binding capacities increased with high values of pH [30].

Equilibrium data, also known as the adsorption isotherm, are the basic requirement for designing an adsorption system. To evaluate the adsorption capacity, we equilibrated IIPs or NIPs (3 mg) with MeHg^+^ solutions (2 mL) in the concentration range of 25–600 µg L^−1^ at pH 8 (Figure 4B). The adsorption value increased with the increase in the concentration of MeHg^+^ in all materials, and it revealed that IIP–MBI–AA (161 µg g^−1^) has higher adsorption capacity than that of non-imprinted polymers NIP–MBI (135 µg g^−1^) and NIP–AA (68 µg g^−1^). The significant difference in adsorption capacities indicates the role of ion imprinting in the adsorbent IIP–MBI. In the same way, IIP–MBT–AA (457 µg g^−1^) and NIP–MBT–AA (416 µg g^−1^) display differences in adsorption capacity, as expected.

The data obtained were converted to Langmuir and Freundlich isotherms, and the related constants are given in Table 3. The Langmuir model is represented in a linear equation form as [49]:(3)$$\frac{C_{e}}{Q_{e}}=\frac{C_{e}}{Q_{m}}+\frac{1}{bQ_{m}}\;,$$
where $C_{e}$ is equilibrium concentration of MeHg^+^ (µg L^−1^), *Q_e_* is the adsorption capacity of MeHg^+^ adsorbed at equilibrium (µg g^−1^) and *Q_m_* and *b* are Langmuir constants, which are related to the adsorption capacity and energy of adsorption, respectively. The plot of *C_e_/Q_e_* against *C_e_* was used to support the Langmuir isotherm.

The Freundlich isotherm is an exponential equation that assumes the sorbate concentration increases together with the adsorbent surface [50]. The linear equation used to study the Freundlich isotherm was
(4)$$\;\log Q_{e}=\log K_{f}+\;\frac{1}{n}\;\log C_{e}\;,$$
where $K_{f}$ is the intercept showing the sorption capacity of the adsorbents, and *n*^−1^ is the slope showing the variation of the sorption with the concentration. The results showed that the Freundlich model fitted (R^2^ = 0.98–0.99) better than the Langmuir model (R^2^ = 0.76–0.84) with the experimental behavior of the sorption process. This suggests the interactions and adsorption occur on multiple layers of the sorbents, which is an expected behavior for non-covalently-imprinted polymers.

To study the kinetic of MeHg^+^ sorption onto IIPs and NIPs, batch experiments were carried out. Figure 4C shows that MeHg^+^ adsorptions reach the maximum capacity after around 120 min for IIP–MBI–AA, NIP–MBI–AA and NIP–AA, and 95 min for IIP–MBT–AA and NIP–MBT–AA. The kinetic data were fitted using the pseudo-first-order and pseudo-second-order kinetic models, described by Equations (5) and (6), respectively [35,51].
(5)$$\log\left(Q_{e}-Q_{t}\right)\;=\;\log Q_{e}-\;\frac{K_{1}}{2.303}t\;,$$
(6)$$\frac{t}{Q_{t}}=\frac{1}{K_{2}Q_{e}^{2}}+\frac{1}{Q_{e}}t\;,\;$$
where *Q_e_* (μg g^−1^) and *Q_t_* (μg g^−1^) are the amounts of adsorbate adsorbed at equilibrium and at time t, respectively. *K*_1_ (min^−1^) and *K*_2_ (g μg^−1^ min^−1^) are the pseudo-first-order and pseudo-second-order adsorption rate constants, respectively. The pseudo-first-order and pseudo-second-order kinetic models were used to model the experimental data. From Table 4, we can conclude that the pseudo-second-order kinetic model (R^2^ ≤ 1) provided a good correlation for the adsorption of MeHg^+^ on IIPs and NIPs compared to the pseudo-first-order (R^2^ << 1) kinetic model. Besides, the theoretical values (*Q_the_*) calculated through the pseudo-second-order model for all polymers were close to the experimental values (*Q_exp_*).

In order to investigate and to compare the selectivity of IIP–MBI–AA and IIP–MBT–AA for MeHg^+^, some metal ions (Hg^2+^, Cd^2+^, Pb^2+^, and Zn^2+^) were chosen as competitive metals to prepare binary mixtures. The ion Hg^2+^ was selected as a potential interferent because it often coexists with MeHg^+^ in wastewater and biological samples. The ion Cd^2+^ was chosen because of its high affinity for thiol groups. The Pb^2+^ and Zn^2+^ metal ions were chosen as competitive ions because of their presence in environmental samples. The adsorption capacity (*Q* in µg g^−1^) and the extraction (*E*) were calculated using the following equations [52]:(7)$$Q=\frac{\left(C_{0}-C_{e}\right)}{m}V\;,$$
(8)$$E=\frac{\left(C_{0}-C_{e}\right)}{C_{0}}\;,\;$$
where *m* is the mass of polymer (mg), *V* is the volume of the solution (mL), and *C_0_* and *C_e_* are the initial and equilibrium concentrations of MeHg^+^ (µg L^−1^), respectively. Figure 4D shows the extraction percentage (obtained by Equation (4)) towards MeHg^+^ and other metal ions. It is evidenced that IIPs and NIPs have selectivity towards mercurial compounds due to sulfhydryl groups in their polymer structures. For MeHg^+^ sorption, the extraction values were 53% and 41% for IIP–MBI–AA and NIP–MBI–AA, respectively. However, higher extraction values were obtained (62% and 60% for IIP–MBI–AA and NIP–MBI–AA) for Hg^2+^ sorption. This suggests that these materials have a higher affinity towards Hg^2+^ ions than MeHg^+^ ions. Differently, MeHg^+^ extraction values were 94% and 92% for IIP–MBT–AA and NIP–MBT–AA, and Hg^2+^ extraction values were 88% and 85% for IIP–MBT–AA and NIP–MBT–AA, respectively.

Table 3 expresses the distribution coefficient (*K_d_*), the selectivity coefficient (*k)* and the impression factor (*I)* values, which could be obtained from the following equations:(9)$$K_{d}=\frac{\left(C_{0}-C_{e}\right)}{m\;Ce}V,$$
(10)$$k=\frac{K_{d}\left(MeHg^{+}\right)}{K_{d}\left(ion\right)}\;,$$
(11)$$I=\frac{k\left(IIP\right)}{k\left(NIP\right)}\;,\;$$

The *K_d_* values of IIP–MBI–AA and IIP–MBT–AA for MeHg^+^ were higher compared to those of the respective NIPs. The *k* for MeHg^+^/Hg^2+^, MeHg^+^/Cd^2+^, MeHg^+^/Pb^2+^ and MeHg^+^/Zn^2+^ in IIP–MBI–AA were 0.89, 20, 5.8 and 35, respectively. On the other hand, the IIP–MBT–AA *k* values were 0.86, 261, 288 and 1510. These results showed that the imprinted polymers retain both species, Hg^2+^ and MeHg^+^, with higher binding capacities. IIP–MBT–AA proved higher affinity than IIP–MBI–AA, and it could be because MBT has two sulfhydryl groups, while MBI only has one sulfhydryl group for complexing MeHg^+^ or Hg^2+^. Taking into consideration the superior extraction efficiency of IIP–MBT–AA, and despite the slightly lower selectivity coefficient, this material was selected for MeHg^+^ sorption in water samples.

### 3.4. Application in Water Samples

The applicability of the IIP–MBT–AA for preconcentration of trace levels of MeHg^+^ was used as a solid phase adsorbent material. It was tested by spiking and recovery experiments on river water and tap water. The water samples spiked with MeHg^+^ at three different environmentally relevant concentrations (25, 50 and 100 µg L^−1^) were treated with the imprinted polymer. The recoveries obtained were between 84% and 95% (Table 5), and the relative standard deviation varied in the range of from 1 to 10%. Recoveries were calculated by the found MeHg^+^ divided per added MeHg^+^. These results indicate the suitability of the present MeHg^+^ imprinted polymer (IIP–MBT–AA) for the quantitative extraction of MeHg^+^ from natural water samples taking into count the complexity of matrix on the river water sample with recovery percentages slightly lower than tap water.

## 4. Conclusions

The computational simulation based on DFT and experimental validations proved to be a powerful tool to design a MeHg^+^ IIP. The theoretical studies showed that MBI, MBT, AA and EtOH were suitable components for the MeHg–IIP synthesis. The performances of the IIP–MBI–AA and IIP–MBT–AA polymers were compared. The effect of pH and adsorption isotherm of imprinted and control polymers revealed differences at pH = 8 with a superior maximum adsorption capacity for the IIP–MBT–AA material. The adsorption isotherm studies indicated that the Freundlich isotherm fitted well the adsorption of MeHg^+^ on the imprinted and control polymers. Additionally, the kinetic study showed follow the pseudo-second-order kinetic equation well in the adsorption process. The selectivity studies showed that the selectivity coefficient *k* values for IIP–MBT–AA were MeHg^+^/Hg^2+^ (0.86), MeHg^+^/Cd^2+^ (261), MeHg^+^/Pb^2+^ (288) and MeHg^+^/Zn^2+^ (1510), revealing that mercurial compounds have high affinity for the recognition sites. The IIP–MBT–AA was successfully applied to concentrate MeHg^+^ ions in water samples with satisfactory results.

## Figures and Tables

**Figure 1 nanomaterials-10-02541-f001:**
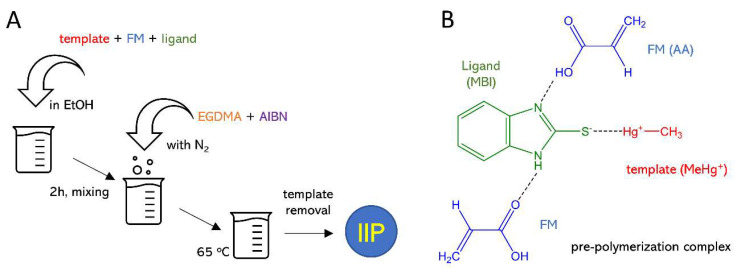
Schematic illustration of the synthesis of ion-imprinted polymers (IIPs) (**A**); pre-polymerization complex with the 2-mercaptobenzimidazole (MBI) ligand (**B**).

**Figure 2 nanomaterials-10-02541-f002:**
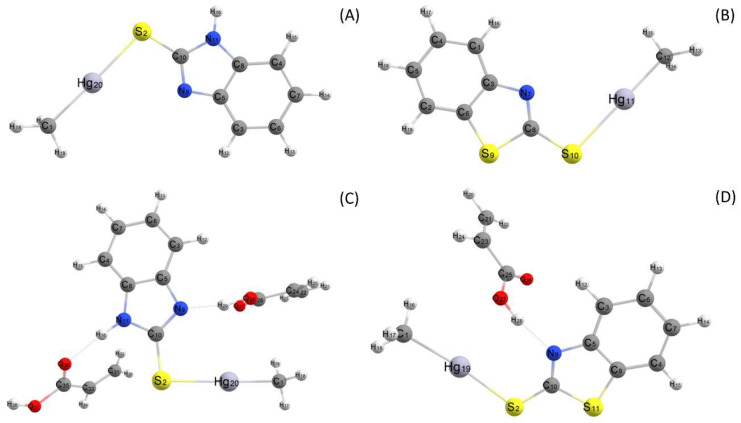
Geometry optimization of methylmercury (MeHg^+^)–MBI (**A**) and MeHg^+^–2-mercaptobenzothiazole (MBT) (**B**) in ethanol (EtOH), and optimized geometries of the pre-polymerization step of MeHg^+^–MBI (**C**) and MeHg^+^–MBT (**D**) complexes with acrylic acid (AA) as the functional monomer (FM).

**Figure 3 nanomaterials-10-02541-f003:**
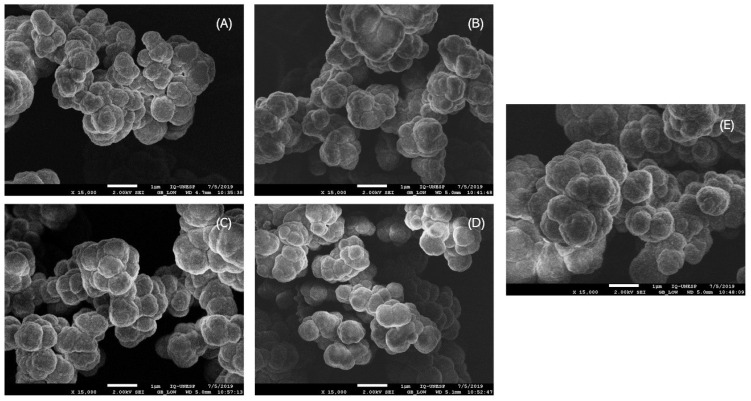
SEM images of (**A**) IIP–MBI–AA, (**B**) non-imprinted polymer (NIP)–MBI–AA, (**C**) IIP–MBT–AA, (**D**) NIP–MBT–AA, and (**E**) NIP–AA; the scale bar is 1 µm in all figures.

**Figure 4 nanomaterials-10-02541-f004:**
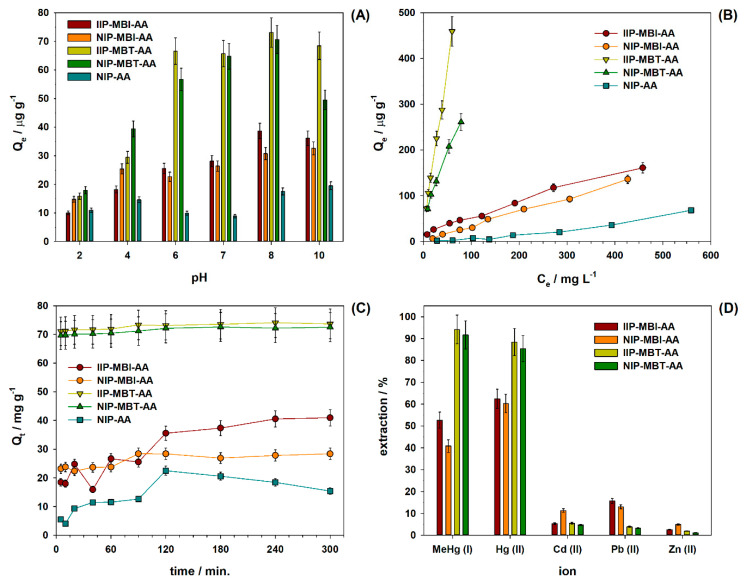
(**A**) Effect of pH on sorption of MeHg^+^ on IIP–MBI–AA, NIP–MBI–AA, IIP–MBT–AA, NIP–MBT–AA, and NIP–AA; 3 mg of polymeric material, [MeHg^+^] = 100 µg L^−1^, volume = 2.0 mL, shaking time = 2 h, room temperature. (**B**) The effect of MeHg^+^ initial concentration on the adsorption of MeHg^+^ on IIP–MBI–AA, NIP–MBI–AA, IIP–MBT–AA NIP–MBT–AA, and NIP–AA; 3 mg of polymeric material, [MeHg^+^] = 25–800 µg L^−1^, pH = 8, volume = 2.0 mL, shaking time = 2 h, room temperature. (**C**) Kinetics of MeHg^+^ adsorption on IIP–MBI–AA, NIP–MBI–AA, IIP–MBT–AA NIP–MBT–AA, and NIP–AA; 3 mg of polymeric material, [MeHg^+^] = 100 µg L^−1^, pH = 8, shaking time = 5–300 min, volume = 2.0 mL, room temperature. (**D**) The selectivity profiles of the IIPs and NIPs for binary mixtures containing MeHg^+^ and potential interferents. In all experiments *n* = 3.

**Table 1 nanomaterials-10-02541-t001:** Solvent effect on binding energy in MeHg ^+^ complexes. The dielectric constant values for water, EtOH, dimethyl sulfoxide (DMSO), dimethylformamide (DMF) and acetonitrile were 78.4, 24, 47, 36.7 and 36.5, respectively.

Complex	Binding Energy/kcal mol^−1^					
	Water	EtOH	DMSO	DMF	Acetonitrile	Vacuum
MeHg^+^–MBI	−10.5	−15.4	−12.2	−13.1	−13.3	−168.5
MeHg^+^–MBT	−7.8	−12.0	−9.2	−10.0	−10.2	−161.5

**Table 2 nanomaterials-10-02541-t002:** Calculated Brunauer–Emmett–Teller (BET) surface area, total pore volume and average pore size.

Material	BET Surface Area/m^2^ g^−1^	Pore Size/nm
IIP–MBI–AA	11	9.5
NIP–MBI–AA	6.8	21
IIP–MBT–AA	5.3	17
NIP–MBT–AA	5.5	11
NIP–AA	3.0	12

**Table 3 nanomaterials-10-02541-t003:** Langmuir and Freundlich isotherm constants.

Material	Langmuir			Freundlich		
	*Q_m_*/μg g^−1^	*b*/L μg^−1^	R^2^	*K_f_*_/_μg g^−1^	*n*^−1^/L μg^−1^	R^2^
IIP–MBI–AA	217.4	0.004	0.76	4.281	0.572	0.98
IIP–MBT–AA	1250	0.009	0.84	13.02	0.866	0.99

**Table 4 nanomaterials-10-02541-t004:** Pseudo-first-order and pseudo-second-order adsorption rate constants.

**Pseudo-First-Order**				
**Material**	***Q_exp_*/μg g^−1^**	***Q_the_*_/_μg g^−1^**	***K*_1__/_min^−1^ × 10^−3^**	**R^2^**
IIP–MBI–AA	40.93	33.23	15.43	0.886
NIP–MBI–AA	28.42	8.72	15.89	0.816
NIP–AA	22.52	14.54	4.606	0.582
IIP–MBT–AA	74.13	2.75	7.369	0.853
NIP–MBT–AA	72.62	3.40	11.28	0.916
**Pseudo-Second-Order**				
**Material**	***Q_exp_*/** **μg g^−1^**	***Q_the_*_/_** **μg g^−1^**	***K*_2__/_mg g^−1^ min^−1^ × 10^−3^**	**R^2^**
IIP–MBI–AA	40.93	44.44	0.678	0.958
NIP–MBI–AA	28.42	28.57	7.239	0.998
NIP–AA	22.52	18.48	2.766	0.941
IIP–MBT–AA	74.13	74.07	15.85	0.999
NIP–MBT–AA	72.62	72.46	14.21	0.999

**Table 5 nanomaterials-10-02541-t005:** Results obtained in recovery tests using two water samples.

Sample	MeHg^+^ Added/µg L^−1^	Found in IIP ^a^/µg L^−1^	Recovery/%
	30.1	25.5 ± 0.4	84.8
river water	60.4	53.7 ± 0.4	88.8
	114.5	103 ± 5	89.6
	22.4	21 ± 2	93.7
tap water	58.2	55 ± 2	95.2
	113.5	101 ± 2	88.5

^a^ Standard deviation for *n* = 3.

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
