# Peer review of "Rational Design of an Ion-Imprinted Polymer for Aqueous Methylmercury Sorption"

_nanomaterials, 2020, doi:10.3390/nano10122541_

Round 1
Reviewer 1 Report
Review: nanomaterials-999191.
Title: Rational design of an ion imprinted polymer for aqueous methylmercury sorption.
In this research-type manuscript, the synthesis and characterization of ion imprinted polymers (MIP) produced from acrylic acid (functional monomers) and 2-mercaptobenzimidazole or 2-mercaptobenzthazole (acting as ion ligands) as well as from ethylene glycol dimethacrylate (cross-linker) was described. The physicochemical characterization of material includes binding parameters fitted into different mathematic models as well as the SEM, FT-IR, TGA and BET measurements. The theoretical investigations of interactions in the prepolymerization complexes were considered. The sorbent was used to prove its capability for separation of methylmercury compound from spiked tap and river water samples. The manuscript has a potential since the selective materials dedicated to analysis of methylmercury compound are important from the environmental and toxicological points of view. However, the analysis of the manuscript revealed several serious drawbacks that are pointed below:
1.The novelty of the manuscript should be clearly demontrated by Authors with respect to recent paper devoted to methylmercury analysis (see: Talanta 2020, in press, doi.org/10.1016/ j.talanta.2020.121841). In the Introduction Section, Authors should discussed the advantages of their study comparing to the above mentioned paper.
2.In the Introduction Section, Authors should refer to other papers devoted to imprinted sorbents towards methylmercury ions (see: Journal of Chromatography A 2017, 1496, 167, RSC Advances 2016, 6, 40100, Talanta 2015, 144, 636, Journal of Chromatography A 2015, 1391, 9, Microchemical Journal 2014, 113, 42). The discussion of above mentioned papers with respect to Author’s study will be appreciated.
3.Introduction Section, in order to reveal full potential of imprinted materials, the recent reviews devoted to imprinting technology and their widespread application shall be described with proper references (see: Chemical Reviews 2019, 119, 94, Nanoscale 2019, 11, 12030, Chemical Society Reviews 2016, 45, 2137, Material Science Engineering C 2017, 76, 1344, Chemical Reviews 2020, 120, 9554). Additionally, recent review related to the topic of Author’s study should be discussed (see: Microchemical Journal 2020, 156, 104886).
4.Designing of sorbent: Figure 1 shall be corrected since the 2-mercaptobenzothiazole ligand does not possess interactions with two monomers, according to theoretical study and Figure 2 (Figure 1 – it is not correct to assign MBI or MBT). Please explain, why the same stoichiometry was used for the synthesis of both sorbents viz., with 2-mercaptobenzimidazole (ligand which interacts with two molecules of acrylic acid monomers) and 2-mercaptobenzothiazole (ligand which interacts with only one molecule of acrylic acid monomer).
5.Removal of ion and purification of the material after synthesis. Please, prove that the ligand which was not covalently bound to the polymer net, was not significantly removed out from the sorbent together with methylmercury template. The FT-IR does not prove the presence of aromatic rings from ligands. In my opinion, Authors should apply another instrumental techniques such as EDS or i.e. elemental analysis to confirm the composition of the sorbent.
6.Please explain the meaning of the sentence (line 186-187): These results show that AA provides stronger IIPs than basic and neutral monomers. Please explain if other monomers, viz. basic or neutral were tested in this study?
7.Results and Discussion. The sorption studies of MeHg2+/Hg2+ system prove that the Hg2+ ions interact (probably by non-specific way) with both sulphur atoms in the 2-mercaptobenzothiazole system and only with one sulphur atom in 2-mercaptobenzimidazole system, explaining higher binding capacity of the material possessing 2-mercaptobenzothazole ligand. Please, comment it.
8.Results and Discussion Section. The BET and TGA analyses shall be discussed more detail. Please, provide data related to micropore area, adsorption/desorption average pore diameter or external surface area since those parameters could be related to the specific adsorption of imprinted material. Please, refer to literature (see: Journal of Separation Science 2019, 42, 1412, Science of the Total Environment 2020, 724, 138151, Polymer Bulletin 2014, 71, 1727 – latter article, which is not exceptional, also refers to the interactions between sulphur atoms in the monomer and adsorbed ions).
9.Experimental Section and Results and Discussion. Authors presents the recovery of metyhlmercury derivative from spiked samples. How the recovery was calculated, what was the solid phase extraction protocol, if the washing step was proceeded, if the contact time of elution was optimized, why the recoveries from river water were lower than those from spiked tap water samples? Please provide sufficient explanations to above mentioned questions.
In my opinion, the manuscript has some potential but serious revision is needed at this stage of evaluation.
Based on above, I recommend major revision of the manuscript.
Author Response
In this research-type manuscript, the synthesis and characterization of ion imprinted polymers (MIP) produced from acrylic acid (functional monomers) and 2-mercaptobenzimidazole or 2-mercaptobenzthazole (acting as ion ligands) as well as from ethylene glycol dimethacrylate (cross-linker) was described. The physicochemical characterization of material includes binding parameters fitted into different mathematic models as well as the SEM, FT-IR, TGA and BET measurements. The theoretical investigations of interactions in the prepolymerization complexes were considered. The sorbent was used to prove its capability for separation of methylmercury compound from spiked tap and river water samples. The manuscript has a potential since the selective materials dedicated to analysis of methylmercury compound are important from the environmental and toxicological points of view. However, the analysis of the manuscript revealed several serious drawbacks that are pointed below:
First, we would like to thank the reviewer for taking the time to evaluate our manuscript. Thanks!
- The novelty of the manuscript should be clearly demontrated by Authors with respect to recent paper devoted to methylmercury analysis (see: Talanta 2020, in press, doi.org/10.1016/ j.talanta.2020.121841). In the Introduction Section, Authors should discuss the advantages of their study comparing to the above-mentioned paper.
The paper mentioned along with others are now cited and differences are discussed. The remarkable advantage of this study, comparing to the above-mentioned paper, is that sulfur compounds act as ion ligands (2-mercaptobenzimidazole and 2-mercaptobenzothiazole). Moreover, these monomers were selected via computational simulation.
- In the Introduction Section, Authors should refer to other papers devoted to imprinted sorbents towards methylmercury ions (see: Journal of Chromatography A 2017, 1496, 167, RSC Advances 2016, 6, 40100, Talanta 2015, 144, 636, Journal of Chromatography A 2015, 1391, 9, Microchemical Journal 2014, 113, 42). The discussion of above-mentioned papers with respect to Author’s study will be appreciated.
The mentioned papers are now cited and discussed in the introduction.
- Introduction Section, in order to reveal full potential of imprinted materials, the recent reviews devoted to imprinting technology and their widespread application shall be described with proper references (see: Chemical Reviews 2019, 119, 94, Nanoscale 2019, 11, 12030, Chemical Society Reviews 2016, 45, 2137, Material Science Engineering C 2017, 76, 1344, Chemical Reviews 2020, 120, 9554). Additionally, recent review related to the topic of Author’s study should be discussed (see: Microchemical Journal 2020, 156, 104886).
As recommended by the reviewer, more references were included in the introduction section, including the cited recent review (Microchemical Journal 2020, 156, 104886).
- Designing of sorbent: Figure 1 shall be corrected since the 2-mercaptobenzothiazole ligand does not possess interactions with two monomers, according to theoretical study and Figure 2 (Figure 1 – it is not correct to assign MBI or MBT). Please explain, why the same stoichiometry was used for the synthesis of both sorbents viz., with 2-mercaptobenzimidazole (ligand which interacts with two molecules of acrylic acid monomers) and 2-mercaptobenzothiazole (ligand which interacts with only one molecule of acrylic acid monomer).
Figure 1 was corrected because only represents 2-mercaptobenzimidazole MeHg+IIP. To clarify the point about stoichiometry: we decided to keep the same reagents proportions to be compared between polymers.
- Removal of ion and purification of the material after synthesis. Please, prove that the ligand which was not covalently bound to the polymer net, was not significantly removed out from the sorbent together with methylmercury template. The FT-IR does not prove the presence of aromatic rings from ligands. In my opinion, Authors should apply another instrumental techniques such as EDS or i.e. elemental analysis to confirm the composition of the sorbent.
In our opinion, selectivity towards MeHg+, and adsorption essays for IIPs and non-imprinted polymers (NIPs) proved that ligand was trapped in polymer net. As expected, characteristic bands associated to aromatic rings from ligands were not observed in spectral profiles because ligands are in small concentration compared to cross-linker (EGDMA) (1:10).
- Please explain the meaning of the sentence (line 186-187): These results show that AA provides stronger IIPs than basic and neutral monomers. Please explain if other monomers, viz. basic or neutral were tested in this study?
Some functional monomers (acid, basic and neutral) were tested by computational simulation. As a consequence of this study, acrylic acid was selected to IIPs synthesis due to the strongest interaction. Computational simulation saves the time and resources to choose the components for IIP synthesis. About this, more information was added to the manuscript.
- Results and Discussion. The sorption studies of MeHg2+/Hg2+ system prove that the Hg2+ ions interact (probably by non-specific way) with both sulphur atoms in the 2-mercaptobenzothiazole system and only with one sulphur atom in 2-mercaptobenzimidazole system, explaining higher binding capacity of the material possessing 2-mercaptobenzothazole ligand. Please, comment it.
These results showed that the imprinted polymers retain both species, Hg2+ and MeHg+ with higher binding capacities. IIP-MBT-AA proved higher affinity than IIP-MBI-AA, probably due to the presence of 2 sulfhydryl groups in MBT, while MBI has 1 sulfhydryl group for complexing MeHg+ or Hg2+. A comment about this was included in the text.
- Results and Discussion Section. The BET and TGA analyses shall be discussed more detail. Please, provide data related to micropore area, adsorption/desorption average pore diameter or external surface area since those parameters could be related to the specific adsorption of imprinted material. Please, refer to literature (see: Journal of Separation Science 2019, 42, 1412, Science of the Total Environment 2020, 724, 138151, Polymer Bulletin 2014, 71, 1727 – latter article, which is not exceptional, also refers to the interactions between sulphur atoms in the monomer and adsorbed ions).
As recommended by the reviewer, The BET and TGA analyses were discussed more detail and suggested references are now cited in the manuscript. The average pore diameter is named and total pore volume was included in the table 2.
- Experimental Section and Results and Discussion. Authors presents the recovery of metyhlmercury derivative from spiked samples. How the recovery was calculated, what was the solid phase extraction protocol, if the washing step was proceeded, if the contact time of elution was optimized, why the recoveries from river water were lower than those from spiked tap water samples? Please provide sufficient explanations to above mentioned questions.
More explanation to the above mentioned questions was included in the Experimental and Results and Discussion sections.
In my opinion, the manuscript has some potential but serious revision is needed at this stage of evaluation.
Based on above, I recommend major revision of the manuscript.
Reviewer 2 Report
See the added file to authors

Author Response
First, we would like to thank the reviewer for taking the time to evaluate our manuscript. Thanks!
Correct:
Finally, the MeHg+-IIP was successfully applied to the quantitative sorption MeHg+ in samples in river and tap water at environmentally relevant concentrations
as
Finally, the MeHg+-IIP was successfully applied to the quantitative sorption of MeHg+ in samples in river and tap water at environmentally relevant concentrations
Corrected as suggested, thank you for the attention.
Correct:
Figure 1. On the right, schematic illustration of the synthesis of IIPs; on the right, the pre-polymerization complex.
as
Figure 1. On the left, schematic illustration of the synthesis of IIPs; on the right, the pre-polymerization complex.
Corrected as suggested, thank you for the attention.
What is n in equation 2?
Corrected as suggested, thank you for the attention.
Give the significance of abbreviations in table S2!
Corrected as suggested, thank you for the attention.
In figure S2 the ordinate for DTA is missing!
Corrected as suggested, thank you for the attention.
Correct:
An exothermic event is observed due to the loss of water at a temperature below 100 °C in the DTA curve. This peak is followed by a continuous exothermic peak which is responsible for the main decomposition of the polymers (282-333 °C).
With:
An endothermic event is observed due to the loss of water at a temperature below 100 °C in the DTA curve. This peak is followed by an endothermic peak which is responsible for the main decomposition of the polymers (282-333 °C).
Corrected as suggested, thank you for the attention.
You are writing that:
On the other hand, IIP-MBT-AA (73 μg g-1) and NIP-MBT-AA (70 μg g-1) display similar adsorption capacity, but in figure 4A it appears that NIP-MBT-AA has 73 μg g-1 and IIP-MBT-AA has 70 μg g-1
The legend in Figure 4(A) was corrected
Correct
The water samples were spiked with MeHg+ at three different environmentally relevant concentrations (25, 50 and 100 μg L-1) were treated with the imprinted polymer
As
The water samples spiked with MeHg+ at three different environmentally relevant concentrations (25, 50 and 100 μg L-1) were treated with the imprinted polymer
Corrected as suggested, thank you for the attention.
I do not see in table 5 that the relative standard deviation varied in the range 0.09-1.41
Corrected as suggested, thank you for the attention.
You have not a in table 5 but under the table you are writing that:
a Standard deviation for n = 3.
It is the right of IIP, we now make it more visible.
Reviewer 3 Report
The aim of this manuscript was to the evaluate the performance of imprinted polymers for the sorption of MeHg+. The manuscript is potentially interesting for the readers of the Journal and can be considered for publication after changes.
- What is the concentration of MeHg+ in the natural waters ? Please discuss it in the Introduction section.
- Materials and Methods. Please clarify how many replicates per each sample were conducted. It should be useful to provide the error of values in Figure 4.
- Please explain the role of pH on sorption capacity. What is the role of materials studied. The materials studied showed different behavior with the increase of pH. Please explain.
- Application on water samples. What is the effect of water matrix on methylmercury recovery? Did authors conduct quality characterization of the two water samples. How the methylmercury was separated from the polymer ?
Author Response
The aim of this manuscript was to the evaluate the performance of imprinted polymers for the sorption of MeHg+. The manuscript is potentially interesting for the readers of the Journal and can be considered for publication after changes.
First, we would like to thank the reviewer for taking the time to evaluate our manuscript. Thanks!
What is the concentration of MeHg+ in the natural waters ? Please discuss it in the Introduction section.
Such information was now included in the introduction section.
Materials and Methods. Please clarify how many replicates per each sample were conducted. It should be useful to provide the error of values in Figure 4.
The experiments were performed in replicates of three, that information is now included along with error bars in Figure 4.
Please explain the role of pH on sorption capacity. What is the role of materials studied. The materials studied showed different behavior with the increase of pH. Please explain.
The pH has a strong influence the sorption capacity of MeHg+, for example, a pH around 6-8 dictates that most of the MeHg+ would have a hydroxide group associated with it, while at a lower pH (2-4), an ionic speciation for MeHg+ is present. A value of pH 8 was selected to perform all the following experiments because the sorption was more efficient in this condition. About this, more information was added to the manuscript in Results and Discussion section.
Application on water samples. What is the effect of water matrix on methylmercury recovery? Did authors conduct quality characterization of the two water samples. How the methylmercury was separated from the polymer ?
To demonstrate the accuracy of the proposed methodology, we have conducted spiking and recovery experiments in water samples, so that we concluded that although the samples matrices are complex, the proposed method provided an efficient way to sorb methylmercury (without the need for collecting information about the matrix). We did not conduct specific characterization of water samples; however, the difference between the recovery values might be attributed to the higher complexity of the river water compared to tap water. Additionally, the MeHg+ in the IIP was separated with a mixture of tiourea 2.5 % w/v and HCl 1 mol L-1 solution following a previously proposed method.
Round 2
Reviewer 1 Report
Review: nanomaterials-999191_R1.
Title: Rational design of an ion imprinted polymer for aqueous methylmercury sorption.
In this revised manuscript, the Authors have made corrections according to referee comments. In my opinion, the manuscript in current form could be considered for acceptance.
Reviewer 3 Report
Accept in present form